# NORKE: A Framework for Normative Rule-Centric Knowledge Graph Construction using Language Models

Eduardo de Souza[1], André Gomes Regino[2,3,4] and Julio Cesar dos Reis[5]

[1]*Organize Meu Condomínio, Brazil*

[2]*Centro de Tecnologia da Informação Renato Archer (CTI), Brazil*

[3]*Centro Universitário de Jaguariúna (UniFAJ), Brazil*

[4]*Centro Universitário Max Planck (UniMAX), Brazil*

[5]*Instituto de Computação, Universidade Estadual de Campinas (UNICAMP), Campinas, Brazil*

## Abstract

Normative documents exhibit a well-defined logical structure but high semantic density, characterized by recurring terminology, cross-references, and hierarchical rule dependencies. These characteristics pose significant challenges for AI systems, which often struggle to retrieve factually grounded and contextually accurate information from such texts. Despite the growing body of research on Legal Knowledge Graphs (KGs), little attention has been paid to structure-oriented KG construction that explicitly targets the extraction of normative rules from canonical legal elements. Existing approaches often emphasize entity extraction or case-based reasoning rather than modeling the document's internal normative architecture. This investigation proposes the NORKE framework, a structure-aware solution for constructing KGs from normative documents within the Civil Law tradition. Our approach leverages the inherent normative hierarchy, Articles, Caput, Paragraphs, Items, and Sub-items to extract and formalize rules as semantically cohesive RDF triples. Our solution employs an ontology-guided extraction process in which a Large Language Model (LLM) assists in transforming structured normative segments into rule representations aligned with the domain ontology. To evaluate our proposed framework, we constructed a benchmark dataset comprising 700 question–answer pairs drawn from real-world normative documents in Portuguese, including a subset of deliberately ambiguous queries curated by humans. Experimental results show that the resulting KG created by applying NORKE supports factually grounded question answering, achieving an overall approval rate of 82.1% while maintaining stable effectiveness across different levels of linguistic complexity. The obtained findings suggest that our rule-centric KG construction approach can improve traceability and factual grounding when retrieving information from normative documents.

## Keywords

legal documents, knowledge graph construction, large language models,

## 1. Introduction

Normative documents such as regulations, legal statutes, contracts, insurance policies, and institutional bylaws play a central role in organizing social life and regulating commercial relationships. They define obligations, permissions, prohibitions, and sanctions that govern interactions between individuals and organizations. However, the interpretation is often complex and ambiguous. Legal norms may evolve over time through amendments, reinterpretations, or temporal updates, creating situations in which different versions of a rule coexist or conflict. As a result, misunderstandings or misinterpretations of normative texts can have significant social and economic consequences.

Automating the processing of normative documents presents substantial challenges for computational systems. Their language is typically dense, technical, and highly structured, often containing nested clauses, cross-references, exceptions, and hierarchical rule dependencies. In many cases, the intended interpretation of a rule depends on understanding its position within a broader normative hierarchy. For instance, in the Civil Law tradition, normative texts are organized into hierarchical structures such as Articles, Caput, Paragraphs, Items, and Sub-items, in which the **caput** establishes the general rule, and subordinate clauses refine, restrict, or contextualize its application.

✉ eduardo@organizemeucondominio.com.br (E. d. Souza); aregino@cti.gov.br (A. G. Regino); jreis@ic.unicamp.br (J. C. d. Reis)

Computational approaches capable of structuring, interpreting, and explaining normative documents, therefore, have significant societal value. By improving transparency, accessibility, and explainability, these systems can help both specialists and non-specialists better understand the rules that govern everyday social and economic interactions. In particular, they can make normative information more accessible to individuals with limited formal education, thereby enabling a broader segment of society to understand the regulations that shape their rights, obligations, and commercial relationships.

Knowledge Graphs (KGs) provide a promising alternative for representing normative knowledge. Normative documents have a formal structure that can be naturally mapped to graph representations, in which rules, entities, and relationships are encoded as classes and properties. By structuring normative information in this way, KGs enable the explicit representation of rule hierarchies, dependencies, and contextual relationships, thereby enabling more interpretable, traceable, and semantically grounded reasoning. However, constructing KGs is a complex and resource-intensive task. It typically requires substantial human expertise, significant time investment, and considerable financial resources. As a result, the manual development of domain-specific KGs often becomes costly and difficult to scale.

Constructing and maintaining KGs for normative content remains challenging. The ingestion pipeline must ensure adherence to the underlying ontology, as even minor inconsistencies can propagate through inference and produce misleading relationships. These requirements underscore an inherent trade-off: while KG offers richer, more interpretable structures, their creation imposes strict constraints on schema design, extraction accuracy, and ingestion fidelity.

This study proposes NORKE (**NOR**mative **K**nowledge **E**xtraction Framework), a comprehensive computational framework for rule-centric KG construction from normative documents, addressing the described challenge by providing and implementing a structured pipeline for constructing KGs from normative natural-language documents.

NORKE combines a lightweight pre-ontology design with LLMs to facilitate rule extraction and triple generation. It uses a pre-ontological scoping phase to define the conceptual backbone of the domain and defines a T-Box that guides the extraction of structured knowledge from normative documents. LLMs are then used to generate consistent subgraphs that adhere to the ontology constraints and are subsequently ingested into a complete RDF KG in accordance with RML-based mapping principles. Our approach reduces both the financial and computational barriers traditionally associated with KG construction in the context under investigation.

Recent research has explored the use of Large Language Models (LLMs) for KG construction and reasoning over domain-specific texts. Zhu *et al.* [1] analyze the capabilities of LLMs for KG construction, showing that while these models perform well as reasoning assistants, their reliability as standalone extractors remains limited. LLM-based KG construction has also been investigated in domains such as e-commerce [2] and biomedicine [3]. In the legal domain, de Martim [4] proposed an ontology-driven representation of normative documents based on FRBRoo, while Domingos *et al.* [5] introduced a dataset for question answering over Brazilian tax law. Although these studies contribute to knowledge extraction and retrieval over structured corpora, they primarily focus on entity–relation extraction or document-level representations. In contrast, our work focuses on the extraction and explicit representation of prescriptive rules as first-class elements within a KG.

A key innovation of the NORKE is the rule-centric construction of subgraphs using the **caput** of each normative article as the primary semantic anchor. In normative texts, the caput encapsulates the document's core rule, defining an obligation, permission, or prohibition, while subordinate clauses, such as paragraphs, provide additional conditions, restrictions, or clarifications. By treating each **caput** as the root of a rule subgraph enriched with its corresponding classes and properties, our solution captures the essence of normative statements in a structured, explainable manner, while accounting for traceable reasoning paths. Such structured reasoning improves traceability and enables AI systems to justify their answers by referencing the specific normative elements that support them.

Our evaluation used a benchmark dataset (constructed in this study) comprising approximately 700 question-answer pairs derived from real-world normative inquiries in Portuguese. The assessment focuses primarily on factual grounding, measuring whether the generated answer is supported by the governing normative rule represented in the generated KG. The primary metric used is the SAF score

[6], which quantifies the extent to which the generated answers are grounded in the retrieved normative evidence. In addition, we employ Natural Language Inference (NLI) [7] to verify that the generated response is logically entailed by the rule and its supporting clauses. Finally, cosine similarity is used to ensure that the generated response preserves the necessary vector-level semantic similarity to the expected answer for this query, thereby confirming the alignment between the retrieved evidence and the produced response.

The experimental results achieved suggest that linguistic complexity is not a primary source of difficulty in this setting. Because the retrieval process is guided by ontology classes and rule-based queries over the graph structure generated, the system remains stable even when the surface form of the question varies. In other words, conversational or imprecise phrasing appears to have limited impact on the system's ability to identify and retrieve the governing normative rule. These results provide preliminary evidence that our solution for generating KG–based representations can improve factual grounding.

The novelty and contributions of our investigation are:

1. Proposes a rule-centric modeling approach in which normative rules are represented as first-class nodes connected to the entities and clauses that define their scope.
2. Introduces a caput-anchored subgraph automatic construction mechanism, where the caput of each legal article document serves as the semantic root of a structured normative subgraph.
3. Combines ontology-constrained LLM extraction with hierarchical legal segmentation, guaranteeing that the language model populates the A-Box while strictly respecting the predefined T-Box.

Our contributions enable the construction of structured normative KGs that preserve both the hierarchical organization and the prescriptive semantics of legal documents.

The remainder of this article is organized as follows: Section 2 presents related work. Section 3 describes our framework, NORKE. Section 4 reports on the evaluation of NORKE. Section 5 discusses our findings, and Section 6 concludes this article and highlights future investigations.

## 2. Related Work

The application of LLMs to KG construction has been extensively investigated. Zhu *et al.* [1] provided a comprehensive evaluation of LLMs for KG construction and reasoning, demonstrating that while some models show strong effectiveness in reasoning tasks, they exhibit limited reliability as few-shot information extractors. Their experiments indicated that LLMs perform better as reasoning assistants than as primary extractors, highlighting the need for approaches that leverage this specialization. Our study extends this perspective by demonstrating that, in the context of normative documents, advanced reasoning-capable LLMs can extract not only entities and relations, but also complex prescriptive rules when guided by a predefined ontology that encodes the structural organization of normative documents.

Recent studies have demonstrated that LLM-based KG construction from text is being explored across several domains. In the e-commerce domain, Regino *et al.* [2] investigated the use of LLMs to automatically generate KGs from product-related textual data, discussing challenges related to information extraction, entity normalization, and domain-specific vocabulary.

In the biomedical and healthcare domain, Mavridis *et al.* [3] evaluated the use of LLMs to support RDF KG construction through ontology mapping in medical contexts, showing promising results in aligning heterogeneous biomedical terminologies. Similarly, in the cybersecurity domain, Huang and Xiao [8] proposed CTIKG, an LLM-powered pipeline for constructing KGs from cyber threat intelligence reports, enabling structured representation of attack patterns, vulnerabilities, and threat actors. These studies illustrated the growing interest in leveraging LLMs to bridge unstructured text and structured semantic representations across diverse knowledge-intensive fields.

For legal and normative domains, de Martim [4] proposed a solution that explicitly models the hierarchical structure inherent in normative documents. His approach introduced a formal model

inspired by FRBRoo (Functional Requirements for Bibliographic Records), distinguishing between: 1 Norms (abstract works), 2 Hierarchical components (titles, chapters, articles), 3 Temporal Versions (the text as it exists at specific points in time), and 4 Language Versions. By modeling normative evolution as precise aggregations of modified components rather than redundant compositions, his solution enables deterministic and context-aware retrieval of legal texts at any historical point in time.

Domingos *et al.* introduced BR-TaxQA-R [5], a dataset designed for Brazilian tax law question answering that combines statutory documents with administrative case law. Their approach relies primarily on textual segmentation and passage retrieval to support LLM-based question answering. In contrast, our proposal focuses on extracting and explicitly representing prescriptive rules from normative documents within a KG, enabling structured reasoning over normative rules rather than over unstructured text fragments.

Although these reviewed studies establish strong foundations for KG construction in formal domains, they primarily focus on extracting entities and relations or modeling document-level structure. A critical gap remains in the identification, extraction, and explicit representation of prescriptive rules, the normative statements that define obligations, permissions, and prohibitions within legal and regulatory documents. Unlike general knowledge extraction, rule extraction requires understanding complex linguistic patterns, normative expressions (*e.g.*, obligations, permissions, prohibitions), and the interplay between rule components (conditions, subjects, actions) and the entities they govern.

Our present solution addresses this gap by extending the KG paradigm to model rules as first-class citizens within the KG. We introduce a specialized Rule node class that captures not only the structural and temporal context of normative provisions, but also generates a textual representation of the rule's semantics and explicitly connects it to relevant entities, conditions, and governed objects. This rule-centric modeling approach enables more granular reasoning over regulatory compliance and normative interpretation, going beyond the capabilities of existing KG construction methods for normative documents.

## 3. NORKE – NORmative Knowledge Extraction Framework

This section presents a sequential and dependency-driven framework for generating RDF triples from normative documents. Each phase builds upon the previous one, forming a cumulative modeling pipeline in which conceptual abstraction precedes formalization, which in turn constrains extraction. Figure 1 presents our proposed framework.

The process is structured into five interdependent phases: 1. Domain scoping (cf. Subsection 3.1); 2. Ontology design and formalization (cf. Subsection 3.2); 3. Normative-aware textual segmentation (cf. Subsection 3.3); 4. Ontology-constrained LLM triple extraction (cf. Subsection 3.4); and 5. RDF ingestion (cf. Subsection 3.5). Subsection 3.6 presents a running example illustrating the solution.

### 3.1. Domain scoping

Figure 1 (A) presents this stage in the proposed framework. This corresponds to the pre-ontological phase, in which a domain expert defines the domain's conceptual boundaries. This stage follows the scope definition and glossary construction steps described by Fernández-López *et al.* [9]At this stage, a domain expert identifies the key concepts, entities, and relationships that characterize the normative domain under analysis, considering:

1. **Comparative analysis of segmented normative corpora.** This is conducted across distinct normative segments rather than treating regulatory texts as a homogeneous corpus. For instance, technical manuals typically codify procedural obligations and operational constraints, whereas internal regulations formalize behavioral rules and governance structures.
2. **Identification of recurrent semantic patterns.** Beyond surface linguistic variation, this aims to identify recurring prescriptive structures (obligations, permissions, prohibitions), actor-

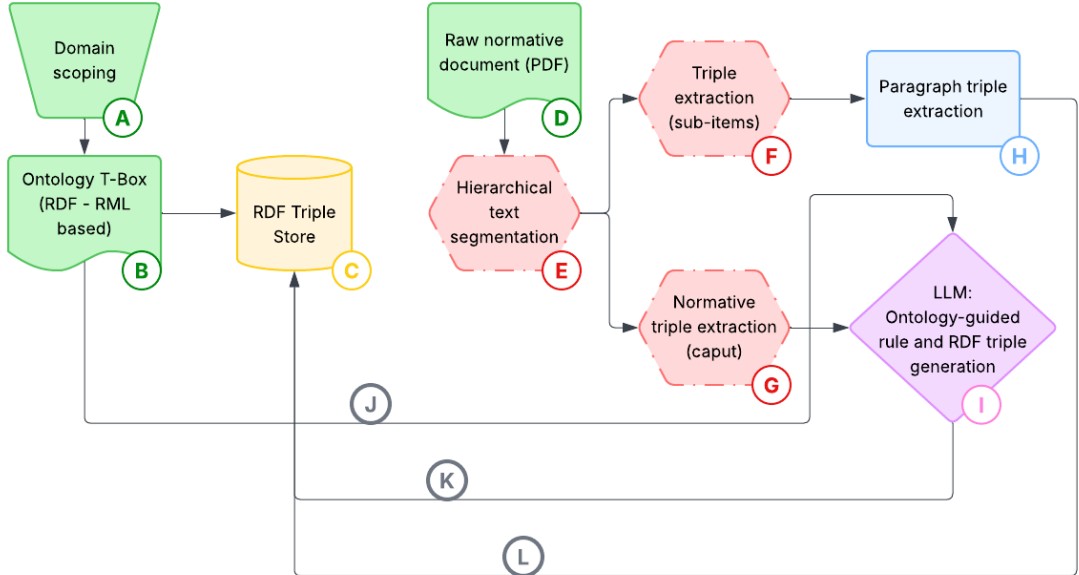

**Figure 1:** NORKE overview. The process begins with domain scoping performed by a human expert (A), leading to the creation of an ontology T-Box expressed in RDF and RML (B) and stored in an RDF triple store (C). Raw normative documents (D) are then processed computationally using hierarchical text segmentation (E). Structural components are extracted as triples, including sub-items (F), while normative rules are extracted from the caput (G) and paragraph-level relations (H). These structured representations are processed by a Large Language Model operating under ontology constraints to generate rule-centered RDF triples (I). The resulting triples are ingested into the triple store in staged steps (J–L), preserving hierarchical relationships.

role configurations, institutional references, and structural hierarchies. To clarify the semantic structure of normative statements, we define a formal representation of a norm.

**Norm.** A norm is a prescriptive statement within a normative document that regulates the behavior of agents by assigning a deontic modality (obligation, permission, or prohibition) to a specific action performed by an agent under defined applicability conditions. In our model, a norm is an abstract entity corresponding to the F1 Work concept of the FRBRoo framework [4]. Formally, a norm $n$ is defined as a tuple:

$$n = (A, Act, C, M)$$

where: $A$ is the set of regulated agents, $Act$ is the regulated action, $C$ is the applicability condition and $M$ is the deontic modality. The modality is defined as:

$$M \in \{Obligation, Permission, Prohibition\}$$

A norm can therefore be expressed as:

$$M(A, Act \mid C)$$

meaning that the modality $M$ applies to agent $A$ performing action $Act$ under condition $C$.

3. **Conceptual clustering of structural elements.** Recurring elements are grouped into five primary conceptual clusters: a) *Core Entities* (objects regulated by the norm, e.g., equipment, spaces, resources); b) *Agents and Roles* (individuals or collectives assuming normative responsibilities); c) *External Institutions* (regulatory bodies, oversight entities, third parties); d) *Normative Artifacts* (documents, rules, articles, clauses); and e) *Auxiliary Contextual Elements* (temporal, conditional, procedural modifiers).

While our detailed experimental analysis focused on condominium regulations and conventions (cf. Section 4), high-level inspection of other normative genres revealed recurring structural patterns, supporting the identification of a shared conceptual backbone across normative discourse. Once defined, the ontology is serialized in RDF and loaded into a RDF Triple Store. This serves as the semantic foundation for subsequent extraction and ingestion processes (cf. Figure 1 (C)).

## 3.2. Manual Ontology design and formalization

For the governance domain, we define a set of top-level ontological axes that categorize entities and operate as semantic constraints for class instantiation. These axes restrict the admissible class space during triple extraction, as illustrated in Figure 1 (B). During the triple extration mechanism in our framework, the prompt design explicitly instructs the LLM to instantiate only predefined classes and properties from the input ontology, thereby prohibiting the creation of new ontological elements, as indicated in Figure 1 (J) and (I). This schema guides the typing of nodes (Subjects and Objects) in RDF triples as follows:

- PhysicalStructure (Classes such as *:Building, :Unit, :CommonArea*);
- PeopleAndRoles (Classes such as *:CondominiumManager, :Resident, :Doorman*);
- ExternalInstitutions (Classes such as *:Supplier, :InsuranceCompany*);
- DocumentsAndRules (Classes such as *:InternalRegulation, :Fine*);
- OperationalActivities (Classes such as *:Maintenance, :Reservation*).

Predicate selection is governed by a semantic control layer termed *Intention*. Intention captures the pragmatic function of the text fragment within normative discourse and determines which property (predicate) connects the nodes in the graph. Three primary intentions are defined for this domain under analysis. Other normative domains may require a different number or type of intentions:

- Regulatory: Encodes permissions, prohibitions, and obligations (e.g., *:permittedIn, :prohibitedIf, :appliesTo*);
- Descriptive: Encodes structural or attributive relations (e.g., *:hasCommonArea, :hasPurpose*);
- Factual/Record: Encodes temporally grounded events (e.g., *:occurredOn, :reservedBy*).

By separating node typing (ontological axes) from predicate selection (intention), we introduce a dual-layer control mechanism that constrains generative behavior at extraction time. This design may reduce the need for extensive human prompt refinement and reduce the risk of hallucinated or structurally inconsistent outputs in the LLM-based generation stage. Such limitations are widely discussed in the context of LLM-based KG construction [1].

The T-Box (terminological component) is manually constructed by a human domain expert prior to any automated extraction in NORKE. All classes and properties are defined a priori and remain fixed throughout the process. At other stages of the framework (cf. Subsection 3.4), the LLM is restricted to populating the A-Box under these constraints, with no permission to modify the ontology.[1]

## 3.3. Normative-aware textual segmentation

A normative document in a Civil Law system is structured hierarchically and textually, typically composed of Articles, Caput, Paragraphs (§), and Items (I, II, III). The *caput* is the main text of a normative article, appearing before any paragraphs or subdivisions. It contains the article's core legal rule and serves as the interpretative basis for all subsequent provisions. To formalize this structural role, we define the caput as the root element of the normative subgraph.

---

[1]Reference:https://anonymous.4open.science/api/repo/ipt-0B01/file/src/ingest/_owl_tbox_v4.ttl?v=8ecc3e3b

**Caput.**  The *caput* is the primary textual component of a normative article and acts as the central interpretative unit for all subordinate clauses. Let:

$$Art = \{art_1, art_2, ..., art_n\}$$

be the set of articles in a normative document. Each article contains exactly one caput. The caput functions as the root node of a normative subgraph:

$$Subgraph(caput_i) = \{caput_i, par_1, par_2, ..., par_k\}$$

where $par_j$ represents subordinate paragraphs.

All rules extracted from the article are anchored to the caput:

$$\forall r \in R, \; source(r) = caput_i$$

This structure ensures that the caput acts as the semantic axis of the normative subgraph.

The normative document is ingested into the framework (cf. Figure 1 (D)). At this stage, the document is prepared for structural processing and segmentation. Our solution uses the Caput as the main node in each subgraph. A unique identifier is assigned to the caput, and this identifier is propagated to all child elements through a parent_id parameter.

At this stage, the document is fully structured, with the Caput acting as the main axis and being connected to each component of the article. An article may consist solely of a caput or include multiple paragraphs, all of which are linked through a shared parent identifier as presented in Figure 1 (E) and (G).

Once normalized, the document is segmented to preserve its semantic integrity. Unlike generic recursive character splitters (*e.g.*, fixed chunk sizes of $1,000$ characters with 200-character overlap), our approach ensures that each structural element (article or paragraph) retains its complete original text exactly as it appears in the source document.

In parallel, the solution generates an additional chunk-based representation within the same document. Complete structural elements (articles or paragraphs) are appended sequentially until an approximate size of 900 characters is reached. Importantly, no article or paragraph is ever split across chunks. Each segment is included in full, even if this slightly exceeds the target size. This approach preserves both semantic and structural coherence, as shown in Figure1 (F) and (G). In the following, we formalize the chunk construction process.

**Chunk.**  Let $D$ be a normative document composed of a sequence of structural elements (*e.g.*, articles, caputs, and paragraphs). A *chunk* is defined as a textual aggregation of complete structural elements that preserves their semantic and hierarchical integrity. Formally, let:

$$SE = \{se_1, se_2, ..., se_n\}$$

be the set of structural elements extracted from $D$. A chunk $c$ is defined as:

$$c = \bigcup_{i=k}^{m} se_i$$

The chunk set is:

$$C = \{c_1, c_2, ..., c_p\}$$

where each chunk preserves the semantic coherence of the original normative structure.

The resulting structured representation preserves both hierarchical metadata and textual content and serves as the input for the ontology-constrained extraction phase (Subsection 3.4).

### 3.4. Ontology-constrained LLM triple extraction

With the ontology in place, each structured representation (*i.e.*, each caput and its associated elements) is processed by an LLM (Figure 1(I)). Rather than performing unconstrained triple extraction, the model operates under a rigid schema defined by a closed set of allowed classes and properties, as presented in Figure 1(J).

For each caput, the model extracts candidate RDF-style triples, explicitly associating every entity and relation with the originating structured representation identifier (`parent_id`). This approach ensures structural consistency, traceability, and compatibility with downstream graph reasoning. For example, if a Caput states that motorcycles are not allowed to park in a designated location, the model produces a triple such as:

$$Motorcycle \rightarrow :notPermittedIn \rightarrow FirstFloorParkingLot$$

The extracted triples are first converted into an intermediate `GraphDocument` representation using the `LLMGraphTransformer` library[2]. Each `GraphDocument` is then transformed into an RDF/Turtle file using the `rdflib` library[3]. The serialization enforces explicit types, namespaces, and property definitions aligned with the T-Box.

Rather than performing generic entity extraction, the LLM is guided to identify the normative actors, regulated objects, and conditions that constitute each rule, explicitly capturing deontic relations (permissions, prohibitions, and obligations) as typed predicates constrained by the T-Box.

These constraints are enforced through a structured prompt that provides the LLM with the complete T-Box definition, the segmented text, and explicit instructions that disallow generating classes or properties outside the predefined schema. The LLM is therefore restricted to populating the A-Box without modifying the underlying ontology.

Beyond extracting explicit triples, the LLM is instructed to instantiate a *:Rule* class element linked to the Caput through the same `parent_id`. The normative rule, even when implicit or distributed across multiple clauses in the original text, is synthesized by the LLM and stored in the `description` field of this element. This representation serves both as a human-readable summary of the normative content and as an additional typed node available for downstream graph reasoning. This design enables SPARQL queries to retrieve, for any given rule, both its textual formulation and the entities to which it is connected.

At the stage of exploring the KG for question answering, the resulting context rule text, together with the linked entities, is provided to the LLM retriever as grounded evidence, ensuring that responses are anchored to explicit normative content.

### 3.5. RDF ingestion

The RDF ingestion stage occurs in two steps. In the first step, the main subgraph corresponding to the Caput is inserted, including all associated nodes (entities, roles, and the *:Rule* instance). In the second step, all paragraphs are inserted as instances of the *:Paragraph* class and connected as child nodes of the corresponding Caput entity through the shared `parent_id`.

This approach to insertion preserves hierarchical integrity and ensures consistency between structural and semantic representations (cf. Figure 1 (K) and (L)).[4]

### 3.6. Running Example

To concretely illustrate the proposed rule-centric construction framework, we introduce a running example based on a simplified excerpt of a condominium regulation.

---

[2]https://pypi.org/project/LLMGraphTransformer/
[3]https://rdflib.readthedocs.io/en/stable/
[4]Reference: https://anonymous.4open.science/api/repo/ipt-0B01/file/src/ingest/output/normativos.ttl?v=5d13df68

***Art. 12.*** *It is prohibited to park motorcycles in the first-floor parking lot.* ***§1.*** *The restriction applies to both residents and visitors.* ***§2.*** *Motorcycles may be parked only in the designated external parking area.*

This excerpt is representative of normative discourse in Civil Law systems: a caput establishing a primary rule, followed by paragraphs that qualify its scope and conditions. Although linguistically compact, it encodes multiple structural and semantic components, including regulated objects (motorcycles), spatial constraints (a first-floor parking lot and an external parking area), actor scope (residents and visitors), and deontic modality (a prohibition combined with a restricted permission).

In the initial stage (1. Domain Scoping), the excerpt is interpreted beyond its surface wording. "Motorcycle" is identified as a core regulated entity, parking areas are categorized as physical structures, residents and visitors are conceptualized as normative roles, and the article itself is classified as a normative artifact. More importantly, the semantic pattern of a prohibition with scoped applicability is recognized as a recurring prescriptive structure within the domain.

These abstractions are then constrained by a predefined ontology (2. Ontology Design and Formalization). Classes such as *Motorcycle*, *ParkingLot*, *Resident*, *Visitor*, and *Rule* are explicitly declared in the T-Box, together with properties encoding normative intentions, such as *notPermittedIn*, *textitappliesTo*, and *allowedOnlyIn*. The ontology functions as a closed schema: during the automatic triple extraction mechanism in NORKE, the solution is restricted to instantiating only predefined classes and properties, preventing uncontrolled relation generation or schema extension.

The document is subsequently parsed according to its legal hierarchy (3. Normative-Aware Segmentation). The caput is assigned a unique identifier and becomes the central node of the subgraph. Paragraphs inherit the same parent identifier, preserving their interpretative dependency on the main rule. Structural boundaries are strictly respected: neither the caput nor any paragraph is fragmented across chunks, thereby ensuring the semantic integrity of each normative unit.

Each structured segment is then submitted to the ontology-constrained language model (4. Ontology-Constrained LLM Triple Extraction). The LLM receives both the segmented text and the complete T-Box specification. From our running example, it generates triples representing that motorcycles are not permitted in the first-floor parking lot and that this restriction applies to residents and visitors, while parking is allowed only in the designated external area.

In addition to these triples, the solution instantiates a dedicated Rule node linked to the caput identifier. This node contains a synthesized textual formulation of the normative statement and connects to regulated entities and roles through typed predicates. By modeling the rule as a first-class entity, the KG captures the unified normative meaning rather than merely isolated binary relations.

This running example demonstrates how a compact normative excerpt is progressively transformed into a structured subgraph centered on a reified rule entity. By aligning abstraction (1), ontological constraint (2), hierarchical parsing (3), controlled extraction (4), and structured ingestion (5), the framework preserves normative semantics and structural dependencies, rather than merely extracting disconnected entities and relations.

## 4. Experimental Evaluation

The objective of this evaluation is to assess the extent to which a KG created via NORKE supports a KG−based retrieval system that generates answers that are: a) factually grounded in the source documents; b) semantically consistent with the expected answers; and c) expressed using appropriate normative language.

**Factual grounding.** Are the answers from input queries supported by evidence in the document? This aspect is evaluated using SAF (Search-Augmented Factuality Evaluator) [6], which measures whether the generated response is grounded in the retrieved context.

**Logical consistency with the ground truth.** Do the generated answers agree with the expected answers? This aspect is evaluated using Natural Language Inference (NLI) [7], which measures whether the generated response entails the expected ground truth.

**Semantic similarity in the legal domain.** Do the generated answers preserve the intended normative meaning? This aspect is evaluated using domain-specific sentence embeddings, which measure semantic similarity within legal Portuguese, ensuring that lexical variation does not obscure normative equivalence.

Subsection 4.1 presents the construction of the ground truth dataset explored in this evaluation. Subsection 4.2 presents a brief description of the real-world documents, the ontologies, and the language models used in this evaluation, and reports the experimental procedures conducted. Subsection 4.3 defines the experimental metrics computed. Subsection 4.4 presents our obtained results.

## 4.1. Evaluation Dataset Construction

A condominium regulation document establishes the rules and procedures governing the use, management, and coexistence within a condominium community. It defines the rights and responsibilities of unit owners, residents, and visitors, including the use of common areas, maintenance obligations, noise and conduct policies, security procedures, and decision-making processes through the condominium association. The document aims to promote orderly administration, protect shared assets, and ensure a respectful and harmonious living environment for all residents while complying with applicable legal and municipal regulations.

The evaluation dataset (for question answering) was constructed from the same condominium regulation document used to build the KG via NORKE in this evaluation. The constructed evaluation dataset was designed to simulate realistic question-answering scenarios encountered in condominium management.

**Question Generation.** The regulation document was processed chapter by chapter, and each chapter served as the contextual source for generating natural-language questions. To approximate realistic user behavior, question generation was performed using an LLM instructed to produce informal and conversational questions inspired by real support interactions between residents and condominium managers. As a reference style, we explored context learning via few-shot learning based on a sample dataset of authentic support inquiries collected from a condominium management platform.

**Prompting Strategy.** For each chapter, the LLM received three inputs: (i) the chapter text, (ii) examples of real user questions, and (iii) instructions to generate questions that reflect typical user language, including incomplete references, narrative descriptions of situations, and everyday vocabulary.

**Reference Answer Definition.** Each generated question was paired with a reference answer (gold standard) manually extracted from the regulation document. The answers range from concise normative statements to longer excerpts that capture relevant conditions, exceptions, or contextual clarifications in the regulation.

The use of the same source document for both KG construction (framework execution in this evaluation) and dataset generation ensures that the evaluation focuses on the system's ability to retrieve and reason over the structured representation of the normative content, rather than on knowledge external to the corpus. The dataset contains 700 question–answer pairs designed to reflect the linguistic variability and ambiguity commonly observed in real-world condominium management interactions.

## 4.2. Inputs and Procedures

The construction of the KG (for the purpose of this evaluation) by executing NORKE combined three complementary elements: i) the normative document as the primary source of truth; ii) the ontology as the semantic schema manually created; and iii) the language model as the rule extraction mechanism.

The resulting Knowledge Graph comprises 66 classes and 40 properties. In terms of instance distribution, the most populated classes are User (775 instances), Rule (353), Chunk (252), CommonArea (63), and PhysicalEntity (52). This distribution indicates that the graph is primarily centered on rule representations and their associated entities.

Rather than storing only the document's textual hierarchy, the NORKE identifies the normative statement expressed in the *Caput* and transforms it into a structured rule linked to the relevant domain

classes. This enables the KG to represent not only where the rule appears in the document, but also what it regulates and which entities are involved. For example, a user query about parking-related situations can be resolved through a semantic traversal such as:

**Query → *Class:Parking* → *Rule:ParkingRule* → *Article:Art12* → *Paragraph:3* → Answer**

This structure allows the system to retrieve all rules associated with a given domain concept, such as `:Parking`, and to expand the contextual evidence provided to the LLM with connected entities such as `:Parkinglot`, `:Bicycle`, and `:Kids`. As a result, questions such as whether children may ride bicycles in the garage can be answered not only through keyword overlap but through explicit links between regulated entities, rules, and documentary evidence.

To support this process, we employed `LLMGraphTransformer` from LangChain with OpenAI o4-mini as the extraction model. The model receives the segmented normative text and the ontology-guided schema and produces RDF triples that instantiate domain entities and explicit rule nodes. This task benefits from language models with stronger reasoning capabilities, since the rule is often implicit in the syntactic structure of the provision rather than explicitly stated as a simple subject – predicate – object relation.

This design improves retrieval precision by making prescriptive knowledge directly queryable through SPARQL and by enabling the downstream LLM to operate over semantically grounded context rather than over isolated text fragments.[5]

To evaluate our solution, we executed the queries in batches. The process begins with query expansion and ontology injection, guided by a 5W3H methodology [10], using GPT-4.1 through the LangExtract framework [6] (cf. Figure 2 (B), (C), and (D)).

When a user submits a natural language question (Figure 2 A), the query is expanded (Figure 2 (B)) to enrich its semantic context. This step reformulates the original question and extracts contextual elements that help align the query with the ontology structure.

Based on this expanded representation, relevant ontology classes are identified (Figure 2 2(D)) using the domain ontology (Figure 2 (C)). This class-extraction stage identifies the ontology classes relevant to the question. This step is performed using the *ChatGPT o4-mini* model, which maps the extracted concepts to the predefined ontology of the KG, as shown in Figure 2 (D) and (E).

The expanded query and extracted classes are then used in the KG retrieval stage, using two queries: (i) a rule-based query that retrieves rules associated with the extracted ontology classes, and (ii) a Lucene-based textual query over the document corpus using the expanded 5W3H query. These two retrieval strategies are executed in parallel: a rule-oriented SPARQL query that retrieves normative rules associated with the identified classes (Figure 2 (E)), and a contextual SPARQL query that retrieves semantically related caput nodes (Figure 2 (F)).

The results of these searches are combined to provide the contextual evidence used by the language model. At this stage, the results from both retrieval paths are provided to the language model for reasoning (Figure 2 (G)), which synthesizes the final grounded answer returned to the user (Figure 2 (H)). In this evaluation, our execution used the *sabia-4* [11] model with temperature 0.1 for final response generation, ensuring deterministic, fact-oriented outputs.

## 4.3. Evaluation Metrics

Figure 3 illustrates how the assessment is conducted using the evaluation metrics explored.

- **SAF (Search-Augmented Factuality)** evaluates whether the generated response is grounded in the retrieved document evidence. It functions as an anti-hallucination safeguard. Responses with $SAF > 0.60$ are considered factually grounded.

---

[5]Reference: https://anonymous.4open.science/api/repo/ipt-0B01/file/mensagens.csv?v=9f797d3f
[6]Reference: https://pypi.org/project/langextract/

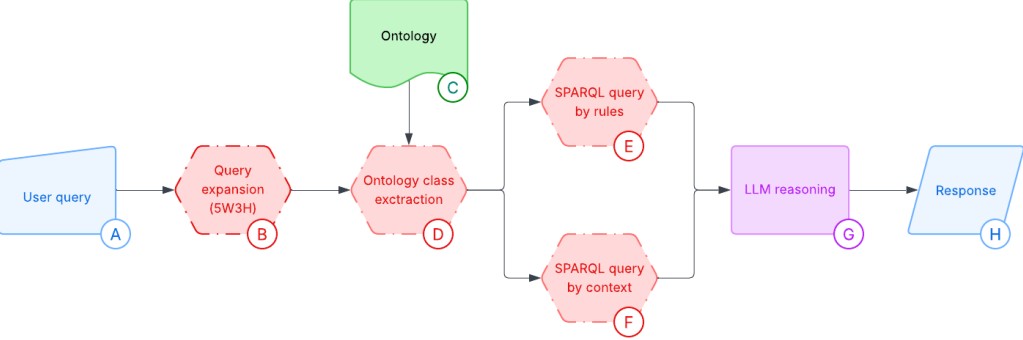

**Figure 2:** The user query is expanded using the 5W3H framework (B). Relevant ontology classes are extracted (D) using the domain ontology (C). Two retrieval strategies are executed in parallel: rule-oriented SPARQL queries over connected rules (E) and contextual retrieval over caput nodes (F). The retrieved evidence is provided to the LLM for reasoning (G), producing the final grounded response (H).

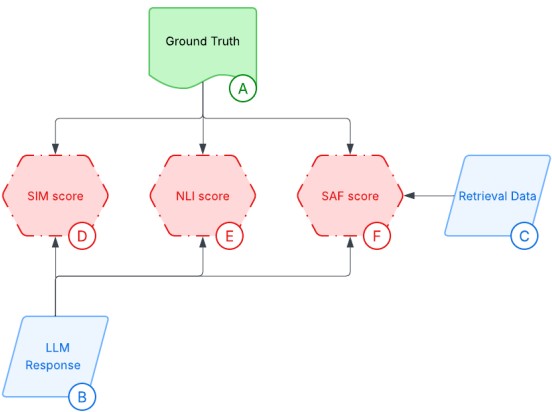

**Figure 3:** The generated LLM response (B) is evaluated against the ground truth answer (A) using complementary evaluation metrics: i) semantic similarity (SIM); ii) natural language inference (NLI); and iii) factual grounding is assessed using the SAF metric, which verifies whether the generated response is supported by the retrieved evidence (C).

- **NLI (Natural Language Inference)** measures whether the generated response logically agrees with the ground truth answer. The model classifies the relationship between the generated answer and the reference answer as entailment, contradiction, or neutral. Only final responses generated with $NLI > 0.75$ are considered strongly aligned with the ground truth.
- **SIM (Semantic Similarity)** evaluates whether the generated answer is semantically similar to the reference (ground truth) answer within the legal domain. Similarity is computed using Legal-BERTimbau embeddings [12] trained on Portuguese legal tasks. Responses with $SIM > 0.60$ are considered semantically consistent with the expected answer.

## 4.4. Experimental Results

The constructed KG via NORKE contains $3,581$ RDF triples and $891$ distinct nodes, including $158$ instances of the *:Rule* class, which represent reified normative statements derived from caput clauses and their associated paragraphs. The KG exhibits a density of $0.0041$ and an average node degree of $7.61$, indicating a sparse but structured topology typical of legally grounded knowledge representations, where relations are semantically constrained rather than densely connected.

Structural segmentation produced $175$ normative subgraphs (Caput-level chunks) with an average of

6.47 nodes per subgraph ($min = 1, max = 17$). Among these, $81\%$ contain at least one rule, with an average of 1.12 rules per subgraph and a maximum of 5 rules in a single structure. These results suggest that the segmentation strategy produces compact, rule-centered clusters that preserve the hierarchical organization of the regulation while enabling localized reasoning over normative statements.

Table 1 summarizes the distribution of evaluation outcomes for the 700 generated answers, categorized into three quality levels: *Approved*, *Needs Review*, and *Rejected*. The majority of responses were classified as *Approved*, comprising 575 instances ($82.1\%$), indicating that most generated answers met the expected quality criteria and were suitable without further intervention. A smaller portion, 64 responses ($9.1\%$), were labeled as *Needs Review*, suggesting that while these answers were partially acceptable, they may require minor adjustments or verification. Finally, 61 responses ($8.7\%$) were classified as *Rejected*, indicating that these outputs did not meet the evaluation standards and should not be used without significant revision. Overall, the results demonstrate a strong predominance of satisfactory outputs, with more than four-fifths of the generated answers meeting the approval criteria.

**Table 1**
Distribution of evaluation outcomes for the 700 generated answers, categorized as **Approved**, **Needs Review**, or **Rejected**.

| Category | Count | Percentage |
|---|---|---|
| Approved | 575 | 82.1% |
| Needs review | 64 | 9.1% |
| Rejected | 61 | 8.7% |
| Total | 700 | 100% |

In our evaluation, questions were stratified into three complexity tiers based on the count of linguistic complexity signals detected. Table 2 presents the results, which show that linguistic complexity does not significantly affect effectiveness. The analysis of 700 questions revealed that linguistic complexity, as measured by the presence of conditional clauses, negations, temporal markers, comparative structures, and multi-part interrogatives, is not correlated with pipeline degradation.

Across the three complexity levels (low, medium, and high), the approval rates remain stable, with values near $82\%$ across all categories. Notably, highly complex questions exhibit a slightly higher approval rate ($84, 1\%$) and the highest mean SAF score ($0.757$), suggesting that the system maintains factual grounding and semantic alignment even in the presence of linguistic complexity signals. Our findings indicate that the proposed approach is robust to variations in question formulation, maintaining consistent effectiveness for both simple and complex user queries.

**Table 2**
Linguistic complexity, as measured by the presence of conditional clauses and multi-part interrogatives, does not correlate with pipeline degradation.

| Complexity Level | N | Approved | Rejected | Mean SAF |
|---|---|---|---|---|
| Low | 476 | 82.1% | 8% | 0.752 |
| Medium | 180 | 81.7% | 11.7% | 0.744 |
| High | 44 | 84.1% | 4.54% | 0.757 |

The results demonstrated that the KG achieves factual grounding and semantic consistency, without noticeable degradation across complexity levels. The structured nature of the KG helps generate answers that can be traced through an explicit reasoning path derived from the KG structure:

**Query** $\rightarrow$ **Class:**`Parking` $\rightarrow$ **Rule:**`Parkingrule` $\rightarrow$ **Article:**`Art12` $\rightarrow$ **Paragraph:**`3` $\rightarrow$ **Answer**

This explicit reasoning path increases transparency by allowing answers to be traced back to the specific rule and clause in the regulation that supports them. Such traceability is valuable in normative

domains where the validity of an answer depends on its ability to reference the governing rule.

## 5. Discussion

Existing solutions struggle to semantically handle facts from normative documents. One of the main advantages of incorporating a KG generated from natural language documents is its ability to make explicit the semantic structure inherent in normative documents. Regulatory texts frequently express permissions, obligations, prohibitions, exceptions, and applicability conditions that depend on relationships between entities. By modeling these relationships explicitly, the KG captures the structural logic of the regulation rather than relying solely on surface textual similarity.

This study highlighted both the potential and the challenges of using KGs to answer normative questions, particularly in scenarios where informal user language is used to query formal regulatory documents in Portuguese. Bridging this gap represents a key challenge for information retrieval systems operating in regulatory environments.

Our findings indicated that linguistic complexity does not significantly degrade system effectiveness, suggesting that conversational or narrative phrasing does not introduce additional difficulty for the retrieval mechanism. Similarly, the average SAF scores remained consistent across the three groups, ranging between $0.744$ and $0.757$, indicating that factual grounding in the retrieved documents is preserved regardless of linguistic complexity. We observed that the system remained stable even when the question's surface form varied. In other words, conversational or imprecise phrasing does not impair the system's ability to identify and retrieve the relevant normative rule. These findings reinforce our underlying hypothesis that our constructed KG can provide robust factual grounding even when queries exhibit the variability typical of real-world user interactions.

Revisiting the evaluation questions introduced in Section 4 allows us to summarize the main findings of this study. First, regarding factual grounding, the results indicate that most generated answers are supported by explicit evidence in the retrieved context. The SAF scores remained high across the evaluated queries, suggesting that the KG-based retrieval pipeline anchors responses to the KG's underlying normative rules. Second, in terms of logical consistency with the ground truth, the NLI evaluation indicates that most generated answers entail the expected reference answers, showing that the system preserves the logical meaning of the governing normative provisions. Regarding semantic similarity in the legal domain, the similarity scores indicate that the generated responses remain aligned with the intended normative meaning of the regulation, even when expressed in different linguistic formulations. These results suggest that the rule-centric KG representation produced by NORKE enables responses that are factually grounded, logically consistent with the governing rules, and semantically aligned with the expected legal interpretation.

Another important observation emerges from the evaluation metrics. Answers in such domains are expected to be short, decisive, and framed in plain language, whereas the gold-standard references often contain lengthy passages embedded in formal legal syntax. The mismatch between these linguistic registers leads to artificially low scores even when the content is correct. Embedding-based metrics, while more tolerant of paraphrasing, still experience limitations. They measure semantic proximity but do not explicitly evaluate factual grounding or logical correctness.

Several avenues for improvement emerge as future work. From the KG's perspective, there is considerable room to expand the ontology to include temporal qualifiers, document versioning properties, and richer representations of exceptions or conditional clauses. Such additions would enable more precise reasoning about when a rule applies and how its applicability changes over time.

## 6. Conclusion

Normative documents contain complex rule structures, exceptions, and applicability conditions that are difficult to interpret when queried. This investigation proposed NORKE, a normative-centric and structure-aware framework for constructing KGs from legal documents. In our approach, entities, rules,

and their hierarchical relationships are explicitly modeled in an ontology, with rules represented as first-class elements connected to domain concepts and normative evidence. The experimental results demonstrated that question answers generated relying on the created KG produce factually grounded answers supported by explicit evidence. The system achieved an overall approval rate of 82.1%, indicating that most responses were both semantically aligned with the expected interpretation and supported by the corresponding normative rule. The explicit representation of rules and document hierarchy enables transparent and traceable reasoning. Our results suggest that rule-centric KG construction can serve as an effective bridge between conversational user queries and the formal structure of normative documents. Future work will explore structural evaluation metrics for rule-oriented KGs and novel KG-based retrieval methods.

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
