# OpenReview forum: "NORKE: A Framework for Normative Rule-Centric Knowledge Graph Construction using Language Models"
_eswc-conferences.org/ESWC/2026/Workshop/KGCW — KGCW 2026_

### Official Review · ~Ibai_Guillén_Pacho1 · 2026-03-30
**KGC in Portuguese Normative Setting**

**Rating:** 7
**Confidence:** 3

**Review:**

## Summary:
The paper introduces NORKE, a framework for constructing Knowledge Graphs from normative documents through a rule-centric pipeline. The approach defines a domain ontology and segments legal texts according to their hierarchical structure. An LLM is then employed to extract RDF triples under strict ontology constraints, generating structured representations of rules that link text elements. The resulting graph enables structured retrieval and question answering grounded in explicit normative rules, demonstrating strong factual consistency and traceability in the evaluation

## Strengths:
The paper presents a well-defined and coherent pipeline for Knowledge Graph construction from normative documents, with a clear emphasis on structure-aware extraction. Its rule-centric modeling, where norms are treated as first-class entities, is a meaningful contribution that goes beyond standard entity–relation extraction. The use of hierarchical document segmentation (e.g., caput and paragraphs) as a semantic backbone is particularly effective for preserving structure during KG construction. Additionally, the integration of ontology-constrained LLM extraction is a strong design choice that improves consistency, traceability, and control over the generated graph, aligning well with key challenges in knowledge graph construction.

## Weaknesses:
The main limitation of the study is its strong focus on a Portuguese normative setting, including the ontology design, the use of a Portuguese LLM, and the evaluation setup, which raises concerns about its generalizability to other legal contexts and whether substantial adaptation would be required. In addition, the approach shows a strong dependence on a manually defined ontology, making the pipeline sensitive to design choices and expert effort. The closed-world extraction constraint, while ensuring consistency, may further reduce coverage by excluding relevant concepts not defined in the schema.

---

### Official Review · ~Lionel_Tailhardat1 · 2026-03-31
**Context graphs for legal documents, but additional details to support the proposal are expected.**

**Rating:** 6
**Confidence:** 4

**Review:**

This work proposes to guide the extraction, by large language models (LLMs), of information from legal documents by integrating, within the models' context, a reference vocabulary and an a priori document structure. The extracted data are then assembled into a knowledge graph for tasks such as document querying and structural analysis.

The paper addresses an important societal need to facilitate the understanding of legal documents and explores an interesting question of controlling an LLM's operation using data derived from a knowledge graph. I particularly appreciate that the approach produces a *context graph* (to borrow a buzzword) for each document. The formalization of the approach is well-handled, the explanations are clear, and the paper is enjoyable to read.

However, a comparison with other works dealing with data extraction from multi-modal documents [1], the use of the GraphRAG technique for LLM-based question answering, or causal relation extraction within documents [2] is missing, which would help better position the approach and the experimental results. A shared dataset appears to be provided alongside the experiments presented. The choice of some experimental parameters is not motivated, and only one LLM (sabia-4) is used in the experimental phase, which limits the interpretability of the results.

Finally, the evaluation relies solely on Q&A generation, similarity calculations, and does not involve expert validations; thus, the cumulative error of NORKE is not necessarily negligible and should be studied to better weigh the performance measures.

Overall, the presented work is interesting but would benefit from further development to fully support all the proposed ideas.

**Specific Remarks and Questions:**

- "Improving transparency, accessibility, and explainability" is highlighted in the introduction (p.2) as a key motivation; however, the authors do not formalize (e.g., as use cases) in which cases text interpretation fails, nor do they clearly specify how NORKE improves the situation in a categorical manner. It is suggested that NORKE enables navigation within texts, but the mechanism of this interaction (a GUI with a graph and hyperlinks?) is not elaborated upon later in the document.

- The paper could improve clarity and conciseness by reorganizing and consolidating certain sections. For example:
  - Elements of "related work" are scattered across multiple sections.
  - A "condominium regulations and conventions" running example could be introduced earlier (e.g., in Section 3) to illustrate the proposals.
  - Figures 1, 2, and 3 are conceptual and are described both in captions and in the text, which could be streamlined.
  - Some references assume implicit knowledge from the reader; it would be helpful to clarify how they support the approach or address specific needs.

- The origin of the "condominium regulations and conventions" dataset and the implementation details of the proposed process are not explicitly stated. Footnotes 1, 4, and 5 refer to a dataset on https://anonymous.4open.science, which corresponds to the presented work. Is there an intention to publish the code and dataset later?

- It is mentioned (p. 7) that the structural elements are "appended sequentially until an approximate size of 900 characters is reached." What is the rationale behind this choice or this limit?

- In Section 3.4, it is stated that constraints are "enforced through a structured prompt" and the use of "explicit instructions." Could the authors present this prompt structure and instructions? Additionally, how do they ensure that constraints are indeed enforced?

- Why not use an LLM for step 1 of the NORKE process?

- Why not employ graph verbalization techniques to compare the answers provided by the LLM to the KG viewed as ground truth?

- Why was T = 0.1 chosen for the sabia-4 model in Section 4.2? What is the impact of other T values on NORKE?

- Why select SAF > 0.60, NLI > 0.75, and SIM > 0.60 in Section 4.3? How do other metric thresholds affect NORKE?

- Could the authors explicitly explain how they measure what they refer to as "linguistic complexity signals" in Section 4.4?

References:
- [1] C. Barboule, B. Piwowarski, et Y. Chabot, « Survey on Question Answering over Visually Rich Documents: Methods, Challenges, and Trends », 2025, arXiv: arXiv:2501.02235. doi: 10.48550/arXiv.2501.02235.
- [2] Youssra Rebboud, Pasquale Lisena, et Raphael Troncy, « Beyond Causality: Representing Event Relations in Knowledge Graphs », in Knowledge Engineering and Knowledge Management, vol. 13514, Oscar Corcho, Laura Hollink, Oliver Kutz, Nicolas Troquard, et Fajar J. Ekaputra, Éd., Cham: Springer International Publishing, 2022, p. 121‑135. doi: 10.1007/978-3-031-17105-5_9.

---

### Official Review · ~Manuel_Lama1 · 2026-04-06
**NORKE: A Framework for Normative Rule-Centric Knowledge Graph Construction using Language Models**

**Rating:** 6
**Confidence:** 4

**Review:**

The goal of the paper is highly relevant to the workshop: it addresses the construction of legal knowledge graphs centered on normative rules from legal documents. Starting from a manually designed ontology, the approach automatically generates instances of predefined classes and properties in order to populate the knowledge graph. In addition, it explicitly identifies normative rules from the main provision (caput) of each article. The framework is positively assessed because it clearly identifies the components that make up the proposed solution, which may facilitate the development and comparison of future approaches aimed at addressing the same problem. However, although the evaluation results are promising, there are several issues that limit the overall assessment of the paper’s contribution to the state of the art.

First, the paper does not make sufficiently clear how normative rules are generated and represented. In this regard, there is some ambiguity about the exact nature of such rules in the graph. It seems that the system creates a :Rule instance containing a synthesized textual formulation of the normative statement and linking it to the relevant entities and roles through typed predicates. However, the article does not explain in enough detail how this rule representation differs from the original caput text, nor how exactly it is exploited when answering queries over the knowledge graph. Even the running example does not make explicit how the rule would finally be encoded within the graph. Clarifying this aspect would help to better assess the paper’s contribution from the perspective of knowledge extraction and representation.

Second, a more detailed description of the instance extraction process used to populate the knowledge graph is missing. The paper states that it relies on a library in combination with an LLM and that the resulting structures are later serialized into RDF/Turtle. However, it does not explain what modifications, adaptations, validation steps, or post-processing operations were necessary to make this extraction process effective in practice. This is an important point, since the extraction of triples from the legal text, and therefore the actual construction of the knowledge graph, is a central component of the overall system performance.

Third, the evaluation dataset is built from the same condominium regulation document used to construct the knowledge graph. As the authors note, this design makes it possible to evaluate the system’s ability to retrieve and reason over the structured representation encoded in the KG. However, this choice also raises concerns about the generalizability and robustness of the approach. In particular, it remains unclear to what extent the system would be able to answer questions about legislative documents when they are formulated externally, under different conditions, or across documents and domains not used in the graph construction process. In other words, the current evaluation setup does not fully assess the robustness of the proposed approach beyond the source document used for KG construction.

Overall, the paper presents an interesting contribution, but a more detailed description of certain aspects of the framework implementation is necessary in order to assess the work’s actual contribution.

---

### Decision · Program_Chairs · 2026-04-09

**Decision:**

Accept

**Comment:**

This paper has been selected for presentation at the KGC workshop. We strongly encourage the authors to consider the reviews whilst revising the paper. Camera-ready instructions will soon follow.